# Epidemiological Study of Foot Injuries in the Practice of Sport Climbing

**DOI:** 10.3390/ijerph19074302

**Published:** 2022-04-03

**Authors:** Paula Cobos-Moreno, Álvaro Astasio-Picado, Beatriz Gómez-Martín

**Affiliations:** 1Nursing Department, University of Extremadura, 10600 Plasencia, Spain; pcmoreno@unex.es (P.C.-M.); bgm@unex.es (B.G.-M.); 2Nursing, Physiotherapy and Occupational Therapy Department, Faculty of Health Sciences, University of Castilla-La Mancha, Real Fábrica de Sedas, s/n, 45600 Talavera de la Reina, Spain

**Keywords:** lower limb, sport medicine, clim

## Abstract

Background. Climbing is a multidisciplinary sport, where the main objective is to reach the highest point of a rock wall or to reach the end of an established route. There are different types of modalities: sport climbing and traditional climbing. The risks and precautions taken with respect to this sport will directly affect the epidemiology of injuries related to its practice. The present study was designed to identify and characterize the most frequent injuries in the feet of climbers and to determine if there is a relationship between the injuries that appear and the time spent practicing the sport. Methods. A total of 53 people were collected, 32 men and 21 women, corresponding to the climbers of the FEXME (Extremadura Federation of Mountain and Climbing). To determine the diagnoses, exploratory tests, classified according to the variables to be studied, are carried out: inspection variables and questionnaire variables. Results. The average number of years of climbing was seven years, and the average number of hours of training per week was 6.6 h. Some type of alterations were presented in 70% of the respondents, and foot pain was present during climbing in 45% of the participants. The *p*-value showed a relationship between years of climbing and the occurrence of chronic foot injuries (*p* = 0.035), however, there is no relationship between the occurrence of injuries and chronological age. Conclusion. We can see that the most frequent injuries in the practice of climbing are claw toes, dermal alterations such as bursitis of the first toe and hallux limitus, followed by hallux valgus. Similarly, only a significant relationship was found between the number of years of climbing and the appearance of foot injuries.

## 1. Introduction

The origins of climbing have little to do with sports. Precisely, the first climbers can be considered the scientists who, during the 18th and 19th centuries, discovered a large part of the most important mountains in the world [1]. Climbing is a multidisciplinary sport, where the main objective is to reach the highest point of a rock face or the end of a set route [2]. Different types of modalities can be differentiated 3–4: sport climbing and traditional climbing, in which the climber reaches the “top” of the route and then descends, with the nuance that in sport climbing fixed anchors will be used while, traditionally, the climber must protect against a fall by fixing the anchors to the rock themself [1,2,3]. Sport climbing as we know it today emerged in the 1980s [4,5]. The existence of climbing walls from the 1960s gave a real boost to the evolution of the discipline. In 1991, the first international climbing competition was held in Frankfurt, Germany [2]. Despite such early beginnings, it was not until 1998 when modalities such as bouldering were officially introduced in the field of international competition, according to the official website of the International Federation of Climbing and Mountain Sports (IFSC). It should be noted that female participation has been present since the first competition in both bouldering and sport climbing [6,7]. Sport climbing has become very popular in recent years, being practiced by people of all kinds and ages. This popularity may be due to factors such as the increase in competitive events or the integration of climbing as an Olympic sport in the Tokyo Games. In climbing, successions of acyclic movements are made that seek to move the body [5], where both the hands and the feet [8] actively participate. The federation of said sport promoted the design and construction of synthetic dams, located in sports facilities that simulate the geographical irregularities of a natural space; what are known today as climbing walls [9].

As for the movements of the lower body, there are different technical gestures whose main objective is to bring the center of gravity closer to the wall. Thus, one of these gestures is that of heeling, which will consist of the support of the heel on a foot hold. On the contrary, in the determined gesture “toe”, pressure will be made against a foot prey using the dorsal part of the foot. Despite the fact that most climbing injuries are detected in the upper extremities, joints such as the knee also receive a high level of stress during the execution of some of the aforementioned technical gestures [7]. It is necessary to consider the graduation of the difficulty of the climb, since this may have a certain relationship with the rate of injuries. In addition, the different difficulties in nature climbing are accessible to any age range [8]. The risks and precautions taken regarding this sport practice will directly affect the epidemiology of injuries related to its practice [10].

It is very important to spend adequate time stretching before and after physical activity, to help prevent the appearance of some of these injuries. Performing a progressive warm-up, where the activity gradually increases, will help strengthen the tendons and muscles. The main effect of warming up is to increase the temperature of the muscles that are going to exercise [11]. This gives these structures greater flexibility and, therefore, less susceptibility to tearing under high stress. Another effect of warming up is to increase awareness of the position and movement of the joints. Due to the fact that, as has been mentioned previously, this factor is of great importance in terms of the risk of injury, warming up acquires great importance in the prevention of these [12]. Other techniques, such as applying a little cold after exercise due to its multiple benefits or bandaging the affected area, can prevent the onset of tendinitis or other typical injury related to climbing [10].

There are types of preventive measures available to avoid the above injuries. The combination of these measures together with adequate knowledge of the mechanisms of injury can considerably reduce the risk of incidence. In this way, certain behaviors will be avoided, in addition to taking the necessary measures in case of discomfort, so as not to further aggravate the damage [7,8,9]. Climbing is a sport of growing popularity [11,12,13], meaning the incidence of injuries related to it can increase, so it is convenient to know the type of injuries that climbers can suffer [12,13]. The lesions can be chronic or acute [13,14]. Currently, traumatic injuries have become rare due to improvements in the material used in this sport. Although, it is important to note that many of the injuries are caused by raising the physical limits and the difficulties of the sections [15]. The boom that climbing has caused, by including it in the Tokyo Olympics as a sport, means that more and more people want to practice this sport, which implies that not knowing the sport well can injure them. Hence, the importance of this study is to assess the possible common injuries in the climber’s foot and to be able to prevent these injuries over time. For the literature, this study would be very interesting due to the scarcity of studies on chronic pathologies in the foot due to the practice of climbing. The literature has focused on the upper limbs, leaving the lower limbs, which are no less important, forgotten. Emphasizing, once again, that the foot is one of the most important parts of the body, it is important to know these injuries in order to prevent them or prevent them from getting worse.

The objective of this study is to identify the most frequent injuries in the feet of climbers and to determine if there is a relationship between the injuries that appear and the time of practice of said sport.

## 2. Methodology

### 2.1. Desing and Sample

A descriptive, cross-sectional and prospective study was carried out. The study population is made up of 53 climbers (*n* = 53): 32 men and 21 women, corresponding to the climbers of the FEXME (Extremadura Federation of Mountain and Climbing), who mainly practice climbing on climbing walls. The FEXME is a representative member of the FEDME (Spanish Federation of Mountain Sports and Climbing).

Inclusion and exclusion criteria were applied to the participating population as shown in Table 1.

### 2.2. Equipment and Procedure

The surveys were administered between February and March 2021, at the Cereza Wall climbing wall in Plasencia (Cáceres). To determine the diagnoses, exploratory tests classified according to the variables to be studied are carried out.

On the one hand, variables were collected by questionnaires: age, sex, years of climbing, training time, place of climbing, shoe size, number of feet, cat model and pain scale 16 (mechanical and visual). 

On the other hand, the variables by inspection are morphostructural alterations (hallux valgus, hallux limitus, claw toes), dermatological alterations (blisters, nail problems, hyperkeratotic patterns, hematomas) and nail alterations.

The diseases considered for the statistical analysis were the most frequently found when climbing: skin, nail, joint disorders and pain. Diagnosis was based on clinical signs and symptoms and was determined by a single examiner. Based on these data, an epidemiological study of foot diseases in climbers was carried out.

### 2.3. Ethical Considerations of the Study

The Bioethics and Biosafety Committee of the University of Extremadura (UEx) has reviewed and approved the evaluation and treatment protocols for the data included in this study. It was approved by the committee on 3 March 2021. It is registered under number 15/2021.

All subjects have informed consent. The study was planned and carried out following the ethical principles of the Declaration of Helsinki. In addition, in the treatment of patient data, only those of the subject involved that were essential for the study were processed, discarding data such as the name or medical history of the patient in order to ensure compliance with Organic Law 3/2018, of 5 December, of Protection of Personal Data and Royal Decree 5/2018, of 27 July, which approves the regulations for their development. In addition, the clinical data handled in this investigation have been treated with the utmost confidentiality and custody of the information in accordance with the provisions of Law 41/2002, of 14 November, Regulatory Basic Patient Autonomy.

All participants voluntarily agreed to participate in the study. The subjects underwent a clinical examination prior to the study in order to apply the inclusion/exclusion criteria. Anthropometric data of the subjects such as age, sex, height and weight were also recorded. 

### 2.4. Data Processing and Statistical Analysis

All data were entered from the paper surveys into an Excel worksheet using Microsoft Excel 2010 (Microsoft, Albuquerque, NM, USA). SPSS software version 21.0 for iOS^®^ (IBM, New York, NY, USA) was used for statistical analysis. 

The descriptive analysis of the data included the calculation of frequency and median. Chi-square was used for the association of variables with 95% confidence, in which a level of *p* < 0.05 was statistically significant confidence. 

## 3. Results

The sample consisted of 32 men and 21 women (60% and 40%, respectively). The average age of the sample was 27.5 with a deviation of 12.83. The means of weight and height are 57.75 kg ± 16.94 and 1.62 m ± 17.10, respectively.

All respondents practice sport climbing in a climbing wall, although 60% also practice rock climbing. The average number of hours of training per week is 6.6 h, with 15 h being the maximum number of hours found and the minimum 3 h. The average number of years climbing is seven years. The maximum number of years is 22, and the minimum is 2. (Figure 1).

Over 70% of the people surveyed have some type of alteration or injury to the foot, of which 42% are men and the remaining 32% are women (Figure 2).

The most frequent digital injuries are claw toes, with 59%, an alteration that caused metatarsalgia in climbers. The most common skin lesions are bursitis of the first toe, with 62% of respondents, being more prevalent in people who used smaller shoes and who had a higher degree of climbing difficulty. In addition, the most frequent joint injury, with 52%, is hallux limitus, compared to 40% of hallux valgus. (Table 2). Another important fact to highlight is the presence of hyperkeratosis in the climber’s feet, with the head of the first toe being the most frequent area of appearance, with 51.3%, followed by the backs of the toes, with 21.6%.

Nail alterations were present in 15% of the climbers, the most common being subungual hematomas and onychodystrophies such as onychomycosis. It is worth emphasizing that the most serious alterations were related to those people who had the greatest degree of difficulty in the practice of climbing. (Figure 2). Ankle sprain is found in only 5% of the participants, infrequent compared to other injuries.

The years of climbing practice have a significant association with the presence of injury (*p* = 0.035), so the more years of climbing practice the greater the probability of injury (Table 3), however, there is no association between the presence of injury attributable to chronological age (*p* = 0.534).

Another relevant data in our study is the presence of pain during the practice of climbing: 45% of the participants affirm that they have pain during the practice, and that this pain is found in the first toe.

## 4. Discussion

Next, and following the same structure established in the presentation of the results, we proceed to discuss them. Thus, in the first place, we will proceed to assess the percentage of subjects who present foot injuries with the practice of climbing, followed by what type of injuries are the most frequent, to finalize whether or not there is a relationship between the years of climbing practice and hours of training with the appearance of these injuries. According to the first stated objective, it has been discovered that the most frequent injuries in the practice of climbing are claw toes, followed by bursitis of the first toe and hallux limitus, finding only a relationship to the appearance of injuries with the years of climbing and age of climbing practice. Being a sport that is gaining popularity, as the number of participants increases there will also be a greater number of injuries that will need to be treated in one way or another [2,3,4,5].

Currently, most of the previous literature focuses on acute injuries related to falls [16,17,18], and there are few studies associated with chronic injuries of the climber’s foot that report on the prevalence of these injuries, hence the aim of this study. There are several studies, but they focus on the upper extremity [19,20] more than the lower, and it is important to highlight that the lower extremity is the most important part of the body in climbing, although the opposite is believed, something that Neuhof [21] already stated, as did Volker [22].

In the present study, 70% of the participants had some type of chronic foot injury, 40% in men and 30% in women. It is important to note that there were 20% more men than women, so women are injured more than men, something that Grønhaug already demonstrated in their studies in 2016 [23] and 2018 [24]. They believed that the reason for this difference was the time of injury may be due to the anatomy of the foot in women versus men, but this is not clear, so more research is needed [24]. Anatomic differences in the ankle may account for the difference in reported ankle and foot injuries. Climbing shoes are made primarily for the male ankle, creating a tighter, more stressful pressure point on the female Achilles tendon. This suggestion is supported by the high prevalence of foot/ankle injuries among route climbers in this study [21]. The most common chronic lesions found in the study were claw toes and hallux valgus, followed by dermal alterations such as hallux valgus bursitis and some nail alterations. These pathologies were also described by Largiad [21] and Schöffl [22] years ago in their studies.

The most common chronic injuries found in the study were claw toes and hallux valgus, followed by skin changes such as hallux valgus bursitis and some nail changes. These pathologies were also described by Largiader [25] and Schöffl [26] years ago in their studies. It is important to highlight the need to know the importance of what type of injuries people have and how to prevent them, since lower extremity injuries were still prevalent and possibly more serious than upper extremity injuries. Largiader also refers to the existence of acute injuries such as ankle trauma and sprains, stating that they are one of the most recurrent injuries in climbers; however, in our study less than 10% had suffered a calcaneal sprain or trauma. In this study, it was observed that the longer the climbing time, the higher the probability of lesions; this fact was also observed by Grønhaug G [23,24] and Neuhof [21] in their studies, respectively. 

There is research that focuses on foot pain during climbing, linking this pain to the use of shoes that do not fit well or are too tight [17,27]. This study confirms these findings and reports a higher prevalence of foot pain during climbing, with 45% of participants experiencing foot pain, which may be attributable to poorly fitting footwear commonly chosen for climbing activity.

Regarding the chronological age of the climber, no significant difference was found, something that Schoff [26] already found, stating that the effects of age have shown different results in different studies, pointing out that it is not longevity that affects the risk of injuries, but the years of practice that take. That is, the longer you have been climbing, the greater the chance of injury.

In this study, it is observed that the longer the climbing time, the greater the probability of the presence of lesions, this fact was also observed by Grønhaug [23] and Neuhof [21] in their studies. Previous findings of a higher injury incidence rate among more experienced athletes were due to more hours of training and more extreme movements when climbing. Therefore, professional and elite climbers are more prone to injury than beginners [27]. According to these authors, the cause of this is due to the fact that there is not enough time for the tendons to adapt between sessions, which causes the microtraumas induced by training to worsen, significantly increasing the risk of injury.

In contrast, as a strength of the study, the novelty of this study for the scientific population stands out, since most of the articles that deal with climbing injuries focus on the upper extremity, and in the case of injuries that affect the lower extremity, focus on acute injuries and not on chronic ones, as is accomplished in this study, as is the identification of the most frequent injuries in relation to the years of climbing practice. These data can serve as a reference to establish common chronic injuries in the climber’s foot through future lines of research.

As for the limitations of the study, although the results obtained are conclusive in terms of the study objectives, larger samples could yield more conclusive results. The heterogeneity between the participants means that the results found should be taken with caution. This, in itself, justifies the implementation of future research.

## 5. Conclusions

We can see that the most frequent injuries in the practice of climbing are claw toes, dermal alterations such as bursitis of the first toe and hallux limitus, followed by hallux valgus. Similarly, a significant relationship was only found between the number of years of climbing and the appearance of foot injuries.

## Figures and Tables

**Figure 1 ijerph-19-04302-f001:**
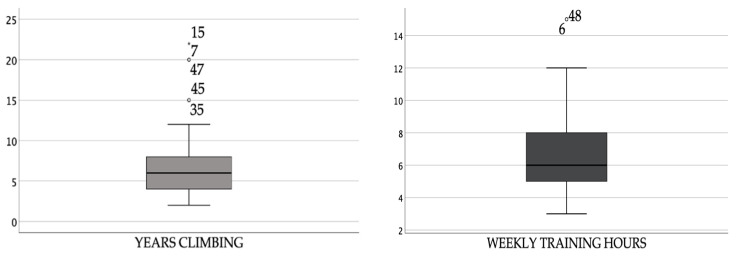
Frequency of years climbing and hours of training. YEARS CLIMBING (the values of the first and third quartiles are 4 and 8, respectively, with a median of 6. The upper whisker is 22, while the lower whisker is 2. Its mean is 7.06); WEEKLY TRAINING HOURS (the values of the first and third quartiles are 5 and 8, respectively, with a median of 6. The upper whisker is 25, while the lower whisker is 3. Its mean is 6.60), extreme cases.

**Figure 2 ijerph-19-04302-f002:**
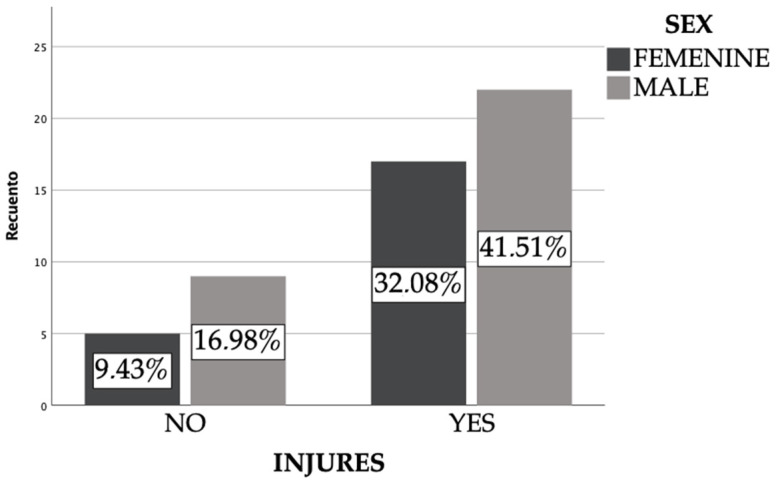
Frequency of injuries in the climber’s foot.

**Table 1 ijerph-19-04302-t001:** Inclusion and exclusion criteria.

Inclusion Criteria	Exclusion Criteria
Climbing for more than year 2 Practice climbing at least 2 days per week Be federated in FEXME *	Having a health problem that alters our sample Having active pain in the lower extremity at the time of sampling

* FEXME (Spanish Federation of Mountain Sports and Climbing).

**Table 2 ijerph-19-04302-t002:** Percentage of climber’s foot injury type.

	Damaged Joint	Dermic Injure	Digital Injure	Nail Injure
%	68	62	68	15
TYPE OF INJURE	Claw (86.7%) Mallet (13.3%)	First finger bursitis (100%)	Limitus (52%) HAV * (40%) Rigidus (8%)	Subungueal Hemtoma (100%)

* HAV (Hallux Abductus Valgus).

**Table 3 ijerph-19-04302-t003:** Contingency table and Chi-square test for the association between the presence of injury and years practicing climbing.

Years Climbing		Less Than 7 Years	7 Years or More	Total
**INJURE?**	YES	38.48%	61.52%	38
	NO	85.72%	14.28%	14
**Chi-square**				**0.035 ***

* Level of significance.

## Data Availability

Data from this study are available from the corresponding author on reasonable request.

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
