# Peer review of "Epidemiological Study of Foot Injuries in the Practice of Sport Climbing"

_ijerph, 2022, doi:10.3390/ijerph19074302_

Round 1

Reviewer 1 Report

The aim of this study was to identify and characterise the most common foot injuries in climbers and to determine whether there is a relationship between the injuries that occur and the length of time spent climbing. The study is novel, but there are some considerations that need to be addressed. 

-Table 1: Removethe points for each item. Include in the table the meaning of FEXME.

-Have the ethical criteria established according to the Helsinki declaration been followed, or would the data protection law alone be correct?

-More data, such as age, weight or height, could be included in the sample. 

-Figures 1,2,3 and 4 should be in better quality and the data inside the columns of the figures are not legible. 

-Revise the bibliography as it should be adapted to the criteria of the journal. 

Author Response

Dear reviewer,
We appreciate your assessment of the manuscript and greatly appreciate major and minor suggestions to enrich it.
As for the main points:
-Table 1: Remove the points of each item. Include in the table the meaning of FEXME: corrected.
-Have the ethical criteria established according to the Declaration of Helsinki been followed, or would the data protection law alone be correct?: clarification is made in lines 125 and 126.
-More data could be included in the sample, such as age, weight or height: information is added on lines 147 and 148.
-Figures 1,2,3 and 4 should be in better quality and the data within the columns of the figures are not legible: the tables are modified following the recommendation of different reviewers.
-Review the bibliography since it must adapt to the journal's criteria: modified according to the indicated criteria.
The modifications made according to your assessment are indicated in the text marked in purple.
We especially appreciate your great review of the article. We have proceeded to review it completely and in detail. We hope it is of your consideration.
Very thankful.

Reviewer 2 Report

General Comments

General Weaknesses

- This section (Methodology) demands format improvements (I recommend to follow the STROBE guidelines -for observational desings- to elaborate it. I recommend to present a structure similar to this one: Study Design, Participants, Protocol/Procedure, Outcomes and Statistical Analysis (pages 3 of 8).  

- Discussion must be improved. I recommend to follow the STROBE guidelines to do it. Overall, the first paragraph must present the main study´s goal and the main study´s results. Subsequently, the whole results of the manuscript must be discussed (these results must be compared with those of the previous studies. In addition, “the underlying mechanisms” of the study´s results must be also presented. Finally, the study´s limitations must be located in the last paragraph of the Discussion (it must be included in Discussion) (pages 6-7 of 8).

General Strengths

- This section is (Results) interesting. It contents useful data for readers, professionals and researchers (pages 3-6 of 8).

- This section is correct (Conclusions) (page 7 of 8).

Major Comments:

Title

Strengths

- Title is pertinent and correct (page 1 of 8).

Abstract

Weaknesses

- Abstract must be elaborated following the present´s report recommendations (page 1 of 8).

Keywords

Weaknesses

- Keywords must be corrected (please, avoid using the same words in the title and in the keywords) (page 1 of 8).

Introduction

Weaknesses

-Despite Introduction is interesting but it must be improved. Firstly, it must be structured in four or five paragraphs which must be focused on specific topics that address the Introduction to the study´s goal. Secondly, the study´s rationaly must be reforced (it is weak at the momento -why does the literature need a epidemiological study of foot injuries in Climbing?-) (pages 1-2 of 8).

Methodology

Weaknesses

- This section (Methodology) demands format improvements (I recommend to follow the STROBE guidelines -for observational desings- to elaborate it. I recommend to present a structure similar to this one: Study Design, Participants, Protocol/Procedure, Outcomes and Statistical Analysis (pages 3 of 8).  

-Statisitcal Analysis.

- This section must be explained in details (being specific, clear and brief) (page 3 of 8).

Results

Strengths

- This section is (Results) interesting. It contents useful data for readers, professionals and researchers (pages 3-6 of 8).

Discussion

Weaknesses

- Discussion must be improved. I recommend to follow the STROBE guidelines to do it. Overall, the first paragraph must present the main study´s goal and the main study´s results. Subsequently, the whole results of the manuscript must be discussed (these results must be compared with those of the previous studies. In addition, “the underlying mechanisms” of the study´s results must be also presented. Finally, the study´s limitations must be located in the last paragraph of the Discussion (it must be included in Discussion) (pages 6-7 of 8).

Conclusions

Strengths

- This section is correct (Conclusions) (page 7 of 8).

References

- This section must be checked it in details. It could contain format mistakes (pages 7-8 of 8).    

Tables and Figures

Weaknesses

-Tables are correct (please, check them, they could content some typos) but there are too many figures. I recommend to include only figures focus on the main  study´s results.

Author Response

Dear reviewer,
We appreciate your assessment of the manuscript and greatly appreciate major and minor suggestions to enrich it.
As for the main points:
- This section (Methodology) requires format improvements (I recommend following the STROBE guidelines -for observational designs- to elaborate it. I recommend presenting a structure similar to this: Study Design, Participants, Protocol/Procedure, Results and Statistical Analysis (pages 3 of 8): Corrections are made on lines 97, 106, 120 and 138.
- The discussion should be improved. I recommend following the STROBE guidelines to do so. In general, the first paragraph should present the main objective of the study and the results of the main study. Subsequently, all the results of the manuscript should be discussed (these results should be compared with those of previous studies). In addition, "the underlying mechanisms" of the study results should also be presented. Finally, the limitations of the study. must be located in the last paragraph of the Discussion (must be included in Discussion) (pages 6-7 of 8): it is modified as indicated between lines 197 to 202. We introduce the limitations at the end of the discussion.
- The summary must be prepared following the recommendations of this report (page 1 of 8): modified lines 11-28.
- Keywords should be corrected (please avoid using the same words in the title and keywords) (page 1 of 8): modified.
The modifications made according to your evaluation are indicated in the text marked in green.
We especially appreciate your great review of the article. We have proceeded to review it completely and in detail. We hope it is of your consideration.
Very thankful.

Reviewer 3 Report

There is a growing interest in climbing sports. The majority of articles focuses on upper extremity injuries, despite there is some evidence from the past studies (doi: 10.5312/wjo.v4.i4.218) that foot is the most commonly injured “climber’s part of the body”. The article challenges prospectively the issue of lower extremity injuries with no such data available. I believe authors provide us with new information in that area of epidemiological knowledge.

Nevertheless, there are some issues that may be improved:

1. Consider exchanging figures to tables or improve size of the fonts as numbers on columns are not clearly visible. Consider providing on figure information with sample total

2. Table 2. There is no information needed in section "NO" as percentage ups always to 100% (option is to provide date as: x of total (%), authors should consider redesigning table to be more clear. 

3. Table 3: Authors should consider providing data with n (%)

4. Author should consider to perform a multivariate analysis (sex, weekly hours, years in training, age, bmi or other factors that were gathered) to include all possible factors influencing the frequency of feet injuries avoiding multiple comparison testing

Minor:

line 86 - citation no brackets

some minor English language  errors, please proof read

Author Response

Dear reviewer,
We appreciate your assessment of the manuscript and greatly appreciate major and minor suggestions to enrich it.
As for the main points:
1. Consider changing figures to tables or improving font sizes as numbers in columns are not clearly visible. Consider providing information on the figure with the sample total: all tables and figures are modified.
2. Table 2. No information needed in "NO" section, as percentage always goes up to 100% (option is to provide date as: x of total (%), authors should consider redrawing table so that be clearer: we have modified the table.
3. Table 3: Authors should consider providing data with n (%): data has been modified by providing the percentage of different cases.
4. The author should consider performing a multivariate analysis (gender, weekly hours, years of training, age, body mass index or other factors that were collected) to include all possible factors that influence the frequency of foot injuries avoiding multiple comparison tests: we carried out these tests but they did not provide very relevant data to the study.
The modifications made according to your evaluation are indicated in the text marked in red.
We especially appreciate your great review of the article. We have proceeded to review it completely and in detail. We hope it is of your consideration.
Very thankful.

Round 2

Reviewer 2 Report

The manuscript has been improved, congratulations for the project. 

Author Response

Dear reviewer,
We especially appreciate your review.
Very thankful.

Reviewer 3 Report

Thank you for your your corections, however there is still few minor point to correct

Figure 1 and 2 – Fonts used in the figures are still small and hardly visible

Figure 1 description – lacking description of box-whisker plot (min-qartile-median etc.)

Table 2 – lacking “%” in mallet

Section: Data availability statement  should be changed

Author Response

Dear reviewer,
We thank you again for your assessment of the manuscript.
As for the main points:
-Figures 1 and 2: the fonts used in the figures are still small and barely visible: fixed.
-Description of figure 1: missing description of the box-whisker plot (min-qartile-median, etc.): added and modified.
-Table 2: missing "%" in deck: added.
-Section: The declaration of data availability must be changed: modified.
We especially appreciate your great review of the article. We hope it is of your consideration.
Very thankful.